# The Development and Evaluation of an Animated Video for Pre- and Postoperative Instructions for Patients with Osteoarthritis—A Design Science Research Approach

**DOI:** 10.3390/geriatrics9010019

**Published:** 2024-02-08

**Authors:** Erik Kylén, Joel Stenholm, Madeleine Johansson, Lena Aggestam, Ann Svensson

**Affiliations:** 1MedFilm AB, Staveredsgatan 20, 461 31 Trollghättan, Sweden; erik.kylen@medfilm.se (E.K.);; 2Department of Adult Psychiatry, NU-Care Hospital, Lärketorpsvägen, 461 73 Trollhättan, Sweden; madeleine.johansson@vgregion.se; 3Department of Engineering Science, University West, Gustava Melins Gata 2, 461 32 Trollhättan, Sweden; 4School of Business Economics and IT, University West, Gustava Melins Gata 2, 461 32 Trollhättan, Sweden

**Keywords:** osteoarthritis, patient instruction, animated video, surgery, design science research

## Abstract

Osteoarthritis (OA) is a condition in the hip or knee joints that develops during a long period of time and sometimes needs hip or knee joint replacement surgery when pain gets too intense for the patient. This paper describes how an animated video for pre- and postoperative instructions for patients with osteoarthritis was designed. The design science research (DSR) approach was followed by creating a web-based animated video. The web-based animated video is used to support surgical departments with education for patients suffering from OA. In the web-based animated video, information about OA surgical treatment and its pre- and post-arrangements was included. The relevance, the rigor, and the design cycles were focused on, with some iterations of and improvements in the animations. Even after implementation, there was a feedback-loop with comments from the surgeons and their patients. Moreover, as more departments will use the web-based animated video, they want to make their special mark on it, so that further changes will be made. This paper presents the design and successful implementation of an animated video for pre- and postoperative instructions for patients with osteoarthritis, tightly linked to the patient journey and the workflow of healthcare professionals. The animated video serves not only as a tool to improve care but also as a basis for further scientific research studies.

## 1. Introduction

Osteoarthritis (OA) is a condition in the hip or knee joints that develops during a long period of time and affects first and mainly articular cartilage [1]. OA is a long-term chronic disease characterized by deterioration. In the final stage, it has affected all the tissues of the joints such as bones, ligaments, joint fluid, and muscles, which results in bones rubbing together and creating stiffness, pain, and impaired movement [2]. This condition can also affect the joints in hands, feet, the spine, and shoulders. However, OA is most common in hips and knees. OA is also the single most common cause of disability in older adults worldwide [3]. It is estimated that 10 to 15 percent of all adults at the age of 60 and above have some degree of OA [4]. Every tenth man and every fifth woman over 60 are estimated to have OA. The condition is not considered apart from the natural aging process, although it is more common among the elderly [5]. Approximately one in four above the age of 45 in Sweden have OA, and in line with an aging and increasingly obese population, the condition is expected to become even more widespread. This will cause increased pressure on healthcare worldwide. However, OA can be treated with physical activity, and anyone with OA can carry out a lot by themselves to improve their health [6]. It is recommended to continue being active, adjusting the activities in accordance with the physical circumstances. However, many patients still require hip or knee joint replacement surgery when pain gets too intense. Studies have demonstrated good outcomes of joint replacement surgery [7]. 

Patients require detailed information about their condition and upcoming surgical procedure, to comprehend the process of pre- and postoperative recommendations; how to prepare before surgery; and how to carry out physiotherapy afterwards [8]. As hospital stays are shortening, the need for information is increasing, as the preparations need to be carried out at home to a greater extent, as well as taking care of their rehabilitation when they return home [9]. The hospitals strive to reduce the number of operations cancelled due to unprepared patients, as this entails costs for unused resources such as surgeons, nurses, as well as operating rooms. Not only is patient satisfaction an important criterion for the overall healthcare service quality, but also for society at large [10].

Healthcare professionals giving medical advice orally has not proven to be a successful method for patient compliance. Neither is handing out written information sufficient, as written instructions can be hard to understand. However, animation and cartoons have been used to improve adherence to instructions, which have shown better patient understanding [11]. Digital tools have also been tested, for example, video-conferencing to provide tele-rehabilitation services, allowing patients to consult with a physiotherapist from their home [12]. The effectiveness of a web-based exercise therapy program tended to improve physical function of the patients [13]. However, digital technologies can be developed to better suit the patients’ needs for information and instructions. Moreover, the increasing focus on efficiency of healthcare is a driving force to find innovative solutions, using design science research methods [14]. Design science research provides an approach to design information systems addressing grand challenges, and in doing so, significantly contributes to healthcare, and society, as it could support development of more than proof of concepts [15]. In this paper, we present a case study about the design and development of a web-based 3D animated video that was developed to support education for patients suffering from OA, using a design science research approach where iterations of group discussions and design of manuscripts were conducted. This study contributes to knowledge on designing 3D animated videos informing patients about pre- and post-operations.

## 2. Materials and Methods

The design science research (DSR) approach was followed by creating a web-based animated video. Two of the authors are specialists in 3D animation and informatics and have been working through the entire design process, from the very beginning in collaboration with healthcare professionals. The third author has been involved in the documentation of the design science research process. The case study presented in this paper is motivated by the healthcare professionals’ desire to improve education for patients. This was made through the introduction of a new and innovative digital application, and the processes behind this. Thus, DSR is a suitable framework to reach this goal. DSR is an iterative process that includes three iterative research cycles [16,17].

The first cycle, the relevance cycle, provided requirements for the application from the very beginning. During this cycle, the research team investigated the problems and opportunities in organizational practice where the resulting artifact should be implemented [18]. In this cycle, our team consulted OA surgeons and other healthcare professionals such as nurses, assistance nurses, and physiotherapists. In the beginning, employees from the hospitals’ communication department were also included. Three group meetings in total were conducted to identify challenges in educating and instructing patients who will be subjected to OA surgery and the following rehabilitation. This cycle also defined the acceptance criteria for the evaluation of the results. The acceptance criteria were defined in a manuscript for the application, with 5 different scenes, each describing «What We See» (Imagery) and «What We Hear» (Dialogue). In each scene, 5 to 25 different parts were describing the acceptance criteria.

The relevance cycle and the design cycle were connected to the rigor cycle in iterations, as literature reviews of the knowledge base of the application domain were conducted. Moreover, expertise was consulted to define the state of the art in the domain and processes of OA surgery, together with patients’ required preparations and rehabilitation activities. The rigor cycle has been used to ensure that the designed application could contribute to research and that the application could achieve an innovative character. The design of the application was grounded in the Technology Acceptance Model (TAM) knowledge base, as well as knowledge collected in other DSR projects [18]. The users’ attitude toward using a digital artifact is dependent on the intention to use it and if the digital artifact is informative and relevant for the intended user group. Therefore, group discussions with design of manuscripts in iterations were used.

The iterative design cycle consisted of two major activities, building and evaluating. Crucial to this design process has been the designer knowledge that is brought into the design activity by the designers [19]. This activity has been tightly related to the designers’ creativity and innovation. This cycle generated design alternatives that were evaluated against the requirements until a satisfactory design was achieved [18]. This cycle generated design alternatives that have been evaluated in discussions with OA surgeons and nurses as representatives from the application domain. A few from the staff of the communication department and a few patients were also included in the iterative design cycle. The evaluating activities were ongoing until a satisfactory design was achieved. Moreover, the evaluations in the design cycle also resulted in additional iterations of the relevance cycle to improve the requirements. This was carried out to increase the application’s utility in practice in the OA surgery domain of increased patient education. Iterative building and evaluating activities are significant characteristics of the DSR framework [19]. This is a novel application, useful for educating patients on what to expect and how to prepare in relation to an OA surgery. Therefore, this study can contribute to the design knowledge [20]. 

## 3. Results: The Rigor Cycle

### 3.1. Related Work

To accomplish the objective to design and develop a 3D animated video to prepare patients for OA surgery, and to teach patients how to conduct the rehabilitation process, we studied OA literature, the literature on technology adoption, and 3D user interfaces. The OA literature informed us on how the hip and knee joints are constructed, why surgery could be necessary, how the patient needs to prepare, how the operation will be conducted, and how the patient should carry out their rehabilitation. It is crucial to understand all these areas to design the application on OA from a patient perspective. We turned to both scientific and popular literature to gain knowledge about the domain. We also had the opportunity to discuss the state of the art in the OA domain with OA surgeons.

### 3.2. Understanding Osteoarthritis and Its Treatments

#### 3.2.1. Osteoarthritis Disease

OA is more common among the elderly but is not a natural part of the aging process. It is true that the cartilage becomes more fragile over the years, just as vision deteriorates and skin loses its tension, but it does not have to lead to OA. OA usually affects older people but can also occur in younger people with rheumatic diseases and injuries, or when the joint is unilaterally strained, for instance, in people who exercise intensely, lift heavy weights, or are overweight.

OA is a condition that occurs throughout the joint, but especially in the cartilage. The joint disease causes the cartilage in the joint to thin out. The pain that arises from OA is related to the cartilage, which acts as a shock absorber in the joint, being too thin or non-existent. This leads to the surfaces of the bones rubbing against each other. The cartilage can be compared to a washing sponge. When the joint is under pressure, joint fluid pushes out of the cartilage and when relieved, the fluid is sucked back. The cartilage has no blood supply, instead it is the joint fluid that is responsible for the nutritional supply. It is important to not load the joints monotonously and constantly for a long time, as the joints need variations. Everyday activities such as walking, cycling, and gardening are good and healthy. However, many people with severe OA have major problems walking.

Being afraid of pain is natural; therefore, it is easy to get scared when an injury hurts. Many people tend to think that the pain is harmful and that the damage will become worse, which is not true. Painkillers are recommended to be able to move properly. There has been a general perception that if one gets OA, the joint probably needs to be replaced in the future. However, only a small amount of people with OA need surgery. To have an operation is a major procedure and full function in the joint may not be regained. Moreover, if an operation is considered to be an option, exercising according to the individual’s specific training program before and after the operation is of vast importance to achieve good results.

The primary healthcare system in Sweden provides OA education for people suffering from this condition, with the goal of providing them with a better quality of life. The education is held by physiotherapists and occupational therapists at primary healthcare centres, enabling patients to learn more about the condition and the treatment options. Moreover, they are also providing help with training programs and advice on how to adapt activities to a level that the individual can manage. Initially, the pain can be relieved with physiotherapy or pain medication. But when the pain becomes so intense that it affects quality of life, a new hip or knee joint is usually the best option. After a new joint prosthesis has been inserted, pain will diminish or almost completely disappear.

The joint replacement surgery is relatively uncomplicated. Spinal anaesthesia is most often administered, and the procedure takes one to two hours. Artificial parts consisting of plastic and metal are used to replace the affected joint. The new joint is usually attached using bone cement.

#### 3.2.2. Osteoarthritis Surgical Treatment and Its Pre- and Post-Arrangements

Even if a joint replacement is considered a routine procedure, there are still various risks present. Therefore, it is important to be carefully prepared as a patient. The patient also needs to inform the physicians if they have any other diseases and whether they are using pharmaceuticals.

Personal hygiene is of great importance, which is why the patient needs to shower twice before surgery using bactericidal soap to avoid infections. Antibiotics are also administered to the patient before surgery. Since blood clots pose a risk, blood thinning medicine is administered from the night before surgery. It is important that the patient gets out of bed and starts walking and exercising the day after surgery. Loosening of the prosthesis is a risk. Therefore, it is important to avoid certain types of exercises that increase potential overloading of the prosthesis, such as running, jumping, and heavy lifting.

Extensive training is required post-surgery to regain strength and mobility. Six to eight weeks of sick leave is to be expected. The first six months after surgery can be difficult because of limited mobility. After approximately one year, full recovery is to be expected.

The life expectancy of a prosthesis is estimated to be 15–20 years. Out of all patients who have undergone joint replacement surgery, 95 percent report good quality of life 10 years post-surgery.

### 3.3. Technology Adoption and 3D Interfaces

The established theoretical model used within areas of the acceptance and adoption of various IT artefacts is the Technology Acceptance Model (TAM) as different versions, TAM1, TAM2 and TAM3, and the Unified Theory of Acceptance and Use of Technology (UTAUT) [21,22,23]. 

Two factors have originally been used in TAM: Perceived Usefulness (PU) and Perceived Ease-Of-Use (PEOU) [24]. PU measures the extent of an individual’s perception to which they believe that using a particular system would be useful for what they want to carry out. PEOU measures how an individual perceives the ease of use of IT, and the degree to which a person believes that using a particular system would be free from effort and without barriers. The direct indicator of the usage of a system is Behavioural Intention to Use (BIU) [25,26]. However, both PU and PEOU have an impact on Attitude Toward Using (A), an individual’s intention to use a specific system, which has an impact on BIU and the actual use of the system [27]. 

We decided to use TAM to evaluate the acceptance of the 3D animated video. Bagozzi [28] claims that various new theoretical models extend the external factors in relation to TAM, which might give the results an unfavourable angle. UTAUT has, for example, 41 independent factors, which make this theory more difficult to use, and TAM3 has a similar design. Vogelsang, Steinhüser, and Hoppe [29] also claim that TAM3 shows significantly worse results compared to TAM and TAM2.

To visualize sequences to inform people, animated 3D models could be used. Three-dimensional visualization facilitates the presentation of subjects that are difficult to present on paper or in 2D [30]. Based on previous experiences, the best choice for starting the design process is that the 3D animators begin by studying the knowledge area by themselves, to define the first version of requirements and evaluation criteria [19]. The 3D animators also claim that the evaluation criteria would be far too medical if the surgeons were involved in defining the first version of the requirements and evaluation criteria. The 3D animated video should be used to inform and teach patients how to handle various pre- and postoperative parts to recover from the operation. Therefore, it is important not to target medically trained persons in this 3D animated video.

The content has been created with the purpose of not showing blood, scalpels, and other surgical instruments to be user friendly for the patients as they play no important role in teaching the patient about OA surgery. It is possible to create a detailed informational video aimed at doctors learning about OA surgery; however, this level of detail is not beneficial to the patient. It is therefore crucial to focus on the patients’ needs during production. Thus, the 3D animation should be easy to understand, yet medically correct and at the same time interesting.

## 4. Results: The Relevance Cycle

To design the application for patients’ education of OA, we first analysed the process of pre- and post-operation of OA surgery, together with intrinsic details of OA and its characteristics. To fully understand the requirements and the evaluation criteria of the application, we have conducted thorough ethnographic research with triangulation. Field observations were carried out by two of the authors, attending OA surgery operations at a hospital in Sweden. Interviews with OA surgeons and group interviews were conducted. The observations were captured in field notes afterwards and the interviews were documented in written notes taken during the interviews. The two group interviews’ discussions were based on different versions of the manuscript for the 3D animation. The results were summarized in a new version of the manuscript containing the sequence of the scenes in the 3D animated video: (1) anatomy, (2) why surgery, (3) before operation, (4) operation, and (5) after operation. A total of 5–25 parts were identified in each of the scenes, as these parts are also defining the evaluation criteria.

The group interview discussions were conducted to obtain insights in the surgical process including pre- and postoperative activities, and to obtain input to be able to conceptualize the first version of the prototype for the 3D animated video. The goal of the relevance cycle was to obtain a detailed understanding before creating the first prototype of the 3D animation as well as the associated requirements and evaluation criteria. Table 1 provides an overview of the operation scene of the first version of the manuscript for the 3D animation, where «what we see» is what is expected to be animated, as not yet analysed, and «what we hear» is what is expected to be told by the spoken narratives in the 3D animated video. Table 2 shows the overview of the very same scene, after it has been iterated in the first group interview discussion.

We can observe that the requirements and evaluation criteria in the scene of the operation are more specified, especially related to the process from B to E in Table 2, compared to Table 1. Also, the Q part is included in Table 2. These changes are based on the group discussion’s focus on the patient. The operation can be seen as divided into three sections in the second version of the manuscript, the first section focusing on what is happening before the cut is made, the second section focusing on what is happening when the surgery is ongoing, and one last section with focus on the completed surgery. This is to inform the patient because they should be well prepared before the surgery, and so the patient should not be surprised by any part of the surgery process.

## 5. Results: The Design Cycle

Together with our insights from the rigor and relevance cycles, our previous experiences in designing 3D animated videos, as well as the technology acceptance literature, we designed two different digital prototypes before the last version of the 3D animated video was released. However, even after the 3D animated video was implemented, we still received feedback on the final result. We followed a user-centred approach following the design requirements in creating the 3D animated video in the design cycle.

### 5.1. First Design Cycle—Animation Exploration

In the first design cycle, we focused on designing the human body as naturally as possible in the animated video. We experimented with both a male and a female human body. We especially focused also on the challenge in finding a balance between showing frightening pictures of body parts and lifelike pictures of the hip joint; see the design process in Figure 1. Insights from the relevance cycle taught us, for example, to avoid showing blood in the animated video. In the first design cycle, we also designed the five scenes with the parts identified in the relevance cycle. In the first design cycle, there is also a focus on finding a balance between the time used for showing the animations and the time it takes for the spoken narrative to relay the corresponding text.

For example, surgeons commented on the length of the incision in the animation. It is of great importance for healthcare professionals to present a realistic scene for their patients. It was also common for the healthcare personnel to want to change the dialogue after they heard the recording—it is sometimes hard to grasp how a sentence sounds beforehand. 

The manuscript from the relevance cycle was used when the video was evaluated by the professionals. They then used the manuscript as a reference to make comments. This evaluation process makes it easier for the 3D animators to make corrections using the feedback given.

### 5.2. Second Design Cycle—Design Validation

In the second design cycle, the different parts in the five scenes were revised, to customize the content to reflect the specific workflow of the clinic in question. The second design cycle also resulted in removing the scalpel from the animated video, and not showing the incision as it looks like in reality. Instead, the incision was shown as a yellow stripe, as in Figure 2.

### 5.3. Third Design Cycle—Validation after Implementation

The third design cycle is an ongoing cycle after the video has been implemented into the daily routines of the clinic. In this cycle, there is a continuous feedback-loop where healthcare personnel and patients from time to time comment on the video content, making the product development a constant process. Two figures, Figure 3 and Figure 4, show the improvements from a previous version to a later version of the video, showing the moment of the artificial hip component that is fitted to the acetabulum. The developed version shows a more pedagogical view of the hip, and the picture is of a better quality.

## 6. Limitations

This study does not include any comparisons with existing methods of patient education. Patients are usually informed on activities and measures included in the course of care in both oral and written documentation. There is also information available on the internet about the operation and its pre- and post-phases, both for the patient and for the relatives and friends. A fourth cycle within the design science research has been suggested by Drechsler and Hevner [31], called the change and impact cycle. This cycle relates a design science project to a wider environmental context, as, for example, a long-term effect and long-term changes or impact. This cycle was not considered within this research study. Therefore, this can be concluded as a limitation of this study, as it cannot conclude something about the results of the 3D animation as patient education. However, the authors are aware of that such 3D animated videos are increasingly used for different surgical interventions.

## *7.* Conclusions and Further Work

It is important to consider each group of professionals working together in a surgical department. The differences become even more clear when implementing the video in another clinic within the same surgical profession. Each department is often keen to change various details in the manuscript to reflect their specific workflow.

Healthcare professionals are often eager to give feedback. However, it seems that different professions give different kinds of feedback, taking pride in their own specific profession but not always seeing the full picture. It is important to consider every branch of healthcare professionals since their expertise needs to be conveyed in the video. During the different design cycles, it became the 3D animators’ responsibility to limit the amount of information and produce a well-balanced video containing all aspects of the patients’ journey. The 3D animated video will continue to evolve, even with other clinics and clients, and it will also be adapted and adjusted to their feedback. This paper has presented the design and successful implementation of an animated video for pre- and postoperative instructions for patients with osteoarthritis, tightly linked to the patient journey and the workflow of healthcare professionals. The animated video serves not only as a tool to improve healthcare but also as a basis for further scientific research studies.

Further research could investigate the fourth cycle of design science, the change and impact cycle, to obtain evidence on the usefulness of such a 3D animated video for pre- and postoperative information for patients with osteoarthritis. It could also be interesting to study whether this 3D animated video made the care more efficient and effective in further research. Such suggestions could include to study if more operations were carried out as planned, or if the patients’ experience of the osteoarthritis operations improved. 

## Figures and Tables

**Figure 1 geriatrics-09-00019-f001:**
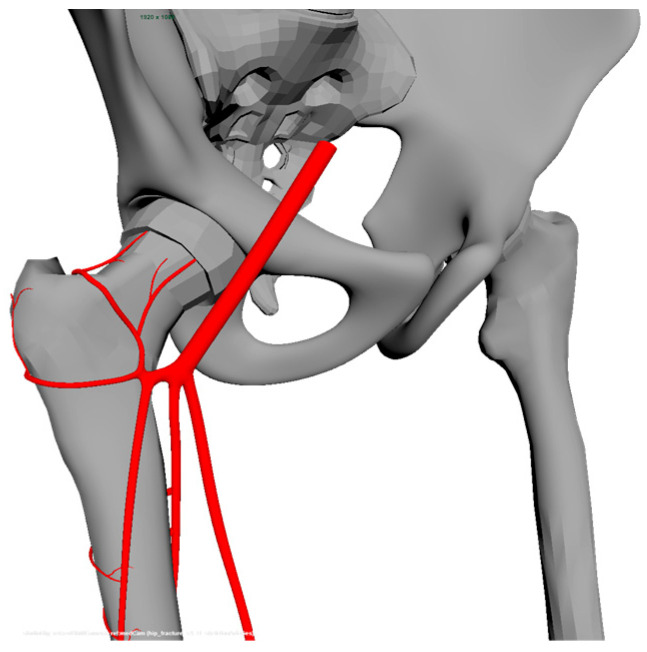
How it looks like in the workflow in the design process before the images are rendered.

**Figure 2 geriatrics-09-00019-f002:**
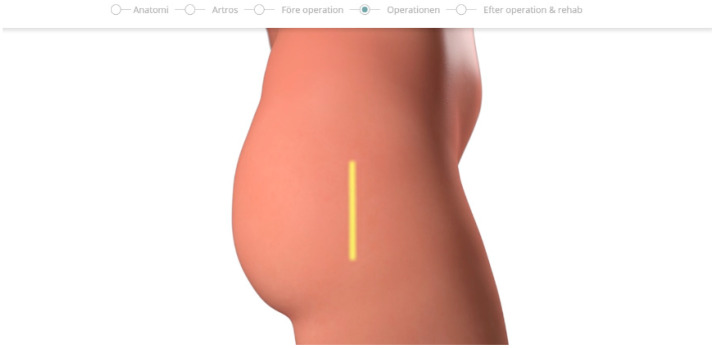
The yellow line shown in the video represents the incision, which is around 15 cm.

**Figure 3 geriatrics-09-00019-f003:**
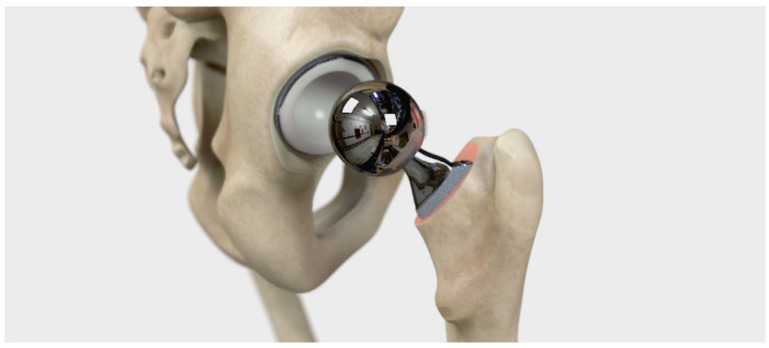
First version of the implemented video, showing the artificial hip component that is fitted to the acetabulum.

**Figure 4 geriatrics-09-00019-f004:**
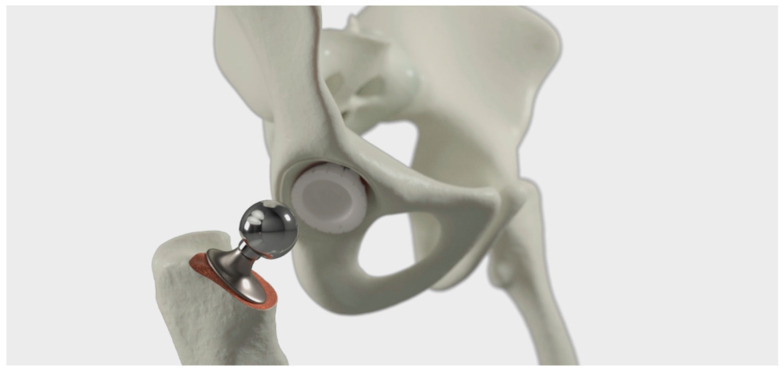
The last version of the video in the section of artificial hip component fitted to the acetabulum, after the feedback-loop comments on the video from healthcare personnel and patients.

**Table 1 geriatrics-09-00019-t001:** The first manuscript for the operation scene. Scene 4: The operation.

Part	What We See (Imagery)	What We Hear (Dialogue)
A		The operation will be performed with either spinal anaesthesia, where only the lower part of the body is numb…
B		…or under general anaesthesia where you are sedated during the procedure.
C		The incision is approximately 15 cm long and will leave a visible scar.
D		The thigh bone is dislocated from the acetabulum.
E		The acetabulum is resurfaced so that the prosthesis will fit.
F		The artificial hip component is then fitted to the acetabulum.
G		A hole is made in the thigh bone to fit the thigh component.
H		The thigh component is then fitted in place.
I		The components are fastened in place with or without the help of bone cement.
J		The artificial ball joint is then placed onto the thigh component to complete the artificial hip.
K		The incision is then closed and the wound is dressed.

**Table 2 geriatrics-09-00019-t002:** The second manuscript for the operation scene. Scene 4: The operation.

Part	What We See (Imagery)	What We Hear (Dialogue)
A		Before surgery, you will receive spinal anaesthesia. You lie on your side and bring your elbows and knees together.
B		When the lower part of your back is washed off with alcohol, the anaesthetist gives you an injection into the spine containing a local anaesthetic through a thin needle.
C		The nerves in the lower part of your body are now anesthetized.
D		The sting itself is hardly felt, but when the anaesthetic strikes, you will begin to feel warm in the legs and you will not be able to move them.
E		During the operation, you can listen to music through headphones. You can also get a sedative so that you fall asleep during the procedure.
F		It is not always possible to administer spinal anaesthesia. Instead, in these cases, we can give you a general anaesthesia.
G		The type of anaesthetic we choose is determined after a consultation with your anaesthetist.
H		The incision is around 15 cm long and will leave a visible scar.
I		The thigh bone is dislocated from the acetabulum.
J		The acetabulum is resurfaced so that the prosthesis will fit.
K		The artificial hip component is then fitted to the acetabulum.
L		A hole is made in the thigh bone to fit the thigh component.
M		The thigh component is then fitted in place.
N		The components are fastened in place with or without the help of bone cement.
O		The artificial ball joint is then placed onto the thigh component to complete the artificial hip.
P		The incision is then closed and the wound is dressed.
Q		A hip replacement takes on average 60–90 min. Longer operating times may occur.

## Data Availability

Available upon request.

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
