# Peer review of "The Development and Evaluation of an Animated Video for Pre- and Postoperative Instructions for Patients with Osteoarthritis—A Design Science Research Approach"

_geriatrics, 2024, doi:10.3390/geriatrics9010019_

Round 1
Reviewer 1 Report
Comments and Suggestions for Authors
I have reviewed the article "Development and Evaluation of an Animated Video for Pre- and Postoperative Instructions for Osteoarthritis Patients - A design science research approach". Here are my suggestions
- I suggest that the design science research approach and how it was applied in the development of the animated video could be more detailed
- decide on whether you use "3D-animated video" or "3D-animated film"
- line 635 - ss is a spelling mistake
- I suggest a section discussing limitations of the study and the animated video development process
- how patients contributed to the video development or their reactions to the final product?
- There is no comparison with existing methods of patient education - elaborate on this
Reviewer 2 Report
Comments and Suggestions for Authors
This article presented scientific novelty and significance. The experimental layout and results were supported authors' hypothesis and the title of this article. Currently, the main text and figures were all kine, therefore revision was not necesary.
Reviewer 3 Report
Comments and Suggestions for Authors
Manuscript titled, Development and Evaluation of an Animated Video for Pre- and Postoperative Instructions for Osteoarthritis Patients. A design science research approach, reports about an animated video for pre- and postoperative instructions for osteoarthritis patients. The paper presents the design and successful implementation of an animated video for pre- and postoperative instructions for osteoarthritis patients, tightly linked to the patient journey and the workflow of health care professionals. The manuscript is well written.
Comments on the Quality of English LanguageMinor editing of English language required
